# Enhancement of Energy-Storage Density in PZT/PZO-Based Multilayer Ferroelectric Thin Films

**DOI:** 10.3390/nano11082141

**Published:** 2021-08-22

**Authors:** Jie Zhang, Yuanyuan Zhang, Qianqian Chen, Xuefeng Chen, Genshui Wang, Xianlin Dong, Jing Yang, Wei Bai, Xiaodong Tang

**Affiliations:** 1Key Laboratory of Polar Materials and Devices, Ministry of Education, Department of Electronic Science, East China Normal University, Shanghai 200241, China; 51191213011@stu.ecnu.edu.cn (J.Z.); 51204700097@stu.ecnu.edu.cn (Q.C.); jyang@ee.ecnu.edu.cn (J.Y.); wbai@ee.ecnu.edu.cn (W.B.); 2The Key Laboratory of Inorganic Functional Materials and Devices, Shanghai Institute of Ceramics, Chinese Academy of Sciences, Shanghai 200050, China; xfchen@mail.sic.ac.cn (X.C.); genshuiwang@mail.sic.ac.cn (G.W.); xldong@mail.sic.ac.cn (X.D.); 3Collaborative Innovation Center of Extreme Optics, Shanxi University, Taiyuan 030006, China

**Keywords:** PZT/PZO, multilayer thin films, electric breakdown field, energy-storage characteristics

## Abstract

PbZr_0.35_Ti_0.65_O_3_ (PZT), PbZrO_3_ (PZO), and PZT/PZO ferroelectric/antiferroelectric multilayer films were prepared on a Pt/Ti/SiO_2_/Si substrate using the sol–gel method. Microstructures and physical properties such as the polarization behaviors, leakage current, dielectric features, and energy-storage characteristics of the three films were systematically explored. All electric field-dependent phase transitions, from sharp to diffused, can be tuned by layer structure, indicated by the polarization, shift current, and dielectric properties. The leakage current behaviors suggested that the layer structure could modulate the current mechanism, including space-charge-limited bulk conduction for single layer films and Schottky emission for multilayer thin films. The electric breakdown strength of a PZT/PZO multilayer structure can be further enhanced to 1760 kV/cm, which is higher than PZT (1162 kV/cm) and PZO (1373 kV/cm) films. A recoverable energy-storage density of 21.1 J/cm^3^ was received in PZT/PZO multilayers due to its high electric breakdown strength. Our results demonstrate that a multilayer structure is an effective method for enhancing energy-storage capacitors.

## 1. Introduction

As electronic components, dielectric capacitors have received extensive investigation from researchers due to their ability to release and store charges [1,2,3]. Dielectric capacitors are the most competitive candidates for current energy-storage electronic devices due to their rapid charge–discharge speed capacity and ultrahigh power density compared to supercapacitors and batteries [4,5]. Extensive efforts have been devoted to enhancing their performance to meet increasing demands related to charge-storage capacitors. Antiferroelectric materials are some of the most prominent dielectrics in electric energy-storage applications given their unique polarization characteristics from linear dielectric and ferroelectric materials [6]. Two key indexes to measure the capability of charge-storage capacitors include recoverable energy-storage density (*W_rec_*) and energy-storage efficiency (η), both of which can be expressed using the following Formulas (1) and (2):(1)$$W_{rec}=\int_{P_{r}}^{P_{m}}EdP$$
(2)$$\eta =\frac{W_{rec}}{W_{rec}+W_{loss}}$$
where *P_m_* and *P_r_* are the maximum and remanent polarization, *E* is the external electric field, and *W_loss_* is energy loss density. Heightening the value of *P_m_*–*P_r_* and the electric breakdown strength (*E_BDS_*) can effectively reduce energy loss and improve the *W_rec_* of energy-storage capacitors. Antiferroelectrics have zero net polarization without an external field due to their reverse-parallel-arranged dipoles. Therefore, antiferroelectric capacitors possess relatively comprehensive energy-storage properties owing to their zero net polarization, even in the case of moderate saturation polarization. It needs to be pointed out that while ferroelectrics have high saturation polarization, they have poor energy-storage characteristics due to their large energy loss and remanent polarization.

Recently, a growing number of researchers have switched focus from simple component material to multicomponent material to further enhance the energy-storage characteristics of dielectric capacitors [7,8,9]. In heterostructures, a layer with high permittivity could share less electric voltage, leading to the decelerating polarization of this layer and improvements in the recovery storage density of the whole heterostructure. For example, BiFeO_3_/BaTiO_3_ bilayers show a significant *W_rec_*, which is improved from 28 to 51 J/cm^3^ [10]. Lu et al. designed a sandwich structure in which the top and bottom layers were BaZr_0.35_Ti_0.65_O_3_ and the middle layer was SiO_2_-doped BaZr_0.35_Ti_0.65_O_3_; this structure exhibited an outstanding energy-storage density of 130.1 J/cm^3^ [11]. Li et al. reported a maximum *W_rec_* of 9.5 J cm^−3^ with an η of 92% at 72 MV/m in NBT-0.45SBT multilayer lead-free ceramic capacitors [9]. The BNBT/2BFO multilayer thin film exhibited energy-storage properties with a recoverable energy density of 31.96 J/cm^3^ and an energy conversion efficiency of 61%, with good thermal stability over a wide temperature range [12]. The insertion of a thin dielectric layer can significantly affect the energy-storage performance of a ferroelectric layer, and Pt/0.5Ba(Zr_0.2_Ti_0.8_)O_3_-0.5(Ba_0.7_Ca_0.3_)TiO_3_/HfO_2_:Al_2_O_3_(HAO)/Au capacitors show an impressive energy-storage density of 99.8 J/cm^3^ and an efficiency of 71.0% [13]. By adding a 30-nanometer low-permittivity Al_2_O_3_ layer in series with a 300-nanometer high-permittivity PZT layer, the voltage in the PZT layer is decreased, which leads to a reduced effective *E* in the PZT layers and a ~640% increase in *W_rec_* [7]. The above multilayer structures, when combined with ferroelectric and dielectric materials, has received extensive attention due to its ability to enhance the electric breakdown strength.

Antiferroelectric material is regarded as one of the most promising energy-storage materials due to its double hysteresis loops with low remnant polarization. For the sake of utilizing both the advantage of ferroelectrics with high *P_s_* and antiferroelectrics with low *P_r_* and high *E_BDS_*, we adopt a similar approach in this paper with PbZr_0.35_Ti_0.65_O_3_(PZT)/PbZrO_3_(PZO) multilayer thin films. A relaxor ferroelectric (Pb_0.92_La_0.08_)(Zr_0.65_Ti_0.35_)O_3_ (PLZT) was used to make a multilayer structure with PZO, and the electric breakdown strength was enhanced dramatically [14]. As a typical antiferroelectric material, PZO is favorable for energy-storage materials with a double hysteresis loop and a nearly zero *P_r_* value, in spite of a low *P_m_* [15,16]. Moreover, PZT is a promising candidate for energy-storage electronic devices owing to its high *P_m_* and strong stability in a large temperature range [17,18]. That said, the low *E_BDS_* is a major disadvantage of PZT, which greatly reduces its application in the field of energy-storage devices. Based on the above consideration, we designed the PZT/PZO multilayers by integrating ferroelectric and antiferroelectric materials together to optimize their energy-storage characteristics using the sol–gel method. Moreover, its multilayer structure is useful for enhancing the breakdown in field strength by taking advantage of the interface resistance and barrier effect of the interface layer [19]. Our results demonstrate that the *E_BDS_* value is 1760 kV/cm for the PZT/PZO multilayer films, which is much larger than for the PZT film (*E_BDS_* = 1162 kV/cm). The *W_rec_* is 21.11 J/cm^3^ at 1760 kV/cm for the PZT/PZO multilayers. Compared with the energy-storage density reported in the literature at the same level of operation voltage, such as 14.8 J/cm^3^ at 1592 kV/cm for PLZT/PZO multilayers and 13 J/cm^3^ at 2400 kV/cm for PZT/Al_2_O_3_/PZT films, our energy-storage density is a little higher under a similar operational electric field; however, our maximum energy-storage density is not larger than others [7,14,20]. It demonstrates that we also need to enhance the electric breakdown strength in the future.

## 2. Materials and Methods

PZT, PZO, and PZT/PZO multilayers were prepared on Pt/Ti/SiO_2_/Si substrates using the sol–gel method [21]. In order to prepare PZT and PZO precursor solutions, an excess of 10 mol% Pb(CH_3_COO)_2_·3H_2_O (AR, China National Medicines Co. Ltd., Shanghai, China) was dissolved in glacial acetic acid solvent (>99.5%, China National Medicines Co. Ltd., Shanghai, China). Acetylacetone (AR, China National Medicines Co. Ltd., Shanghai, China) was added as a chelating agent and refluxed at 80 °C for 30 min. After that, zirconium (IV- butylat Zr(OC_4_H_9_)_4_ (80 wt% in 1-butanol, Sigma-Aldrich Chemical Co. Ltd., Darmstadt, Germany) and titanium (IV isopropoxide C_12_H_28_O_4_Ti (97 wt%, Sigma-Aldrich Chemical Co. Ltd., Darmstadt, Germany) were added according to the stoichiometric ratio in PZO and PZT. Additionally, then, the solution was refluxed at 80 °C for 1 h. Then, we added an appropriate amount of acetic acid to the above solution to obtain a precursor solution with a concentration of 0.1 M. Each PZT layer and PZO layer was spin coated at 5000 rpm for 30 s. After one layer of spin coating, annealing was adopted layer by layer. Every wet layer was burned at 200 °C for 120 s and then pyrolyzed at 400 °C for 120 s; they were eventually annealed at 650 °C for 180 s. These processes were repeated a number of times to attain the required thickness. Finally, three types of film, pure PZT, PZO, and PZT/PZO multilayers, were prepared on the Pt(111)/Ti/SiO_2_/Si substrates, as shown in Figure 1a–c, respectively.

The crystal structure of the three specimens was measured using X-ray diffractometer (XRD, Panalytical Emptrean S3, Almelo, Netherlands). To obtain more morphological information, cross-section images of films were characterized using a commercial scanning electron microscope (SEM, Zeiss GeminiSEM 450, Oberkochen, Germany). The P-E properties were tested using a ferroelectric measurement system (aixACCT TF Analyzer 3000, Aachen, Germany). Dielectric features, including frequency and DC bias external field of the samples, were measured using a highly accurate LCR meter (Agilent E4980A, Palo Alto, CA, USA) with a signal driving voltage of 0.2 V. Current density was measured using a computer-controlled system electrometer (Keithley 6517B, Cleveland, OH, USA).

## 3. Results and Discussion

The XRD patterns of these three films are given in Figure 2a. It can be found that the three types of samples are randomly oriented polycrystalline with a single perovskite structure and no impurity phase. All of the thin films exhibit (111) a preferred orientation and a pseudo-perovskite structure, probably led by the Pt-buffered layer with (111) orientation. Compared with the PZT and PZO films, the PZT/PZO multilayer films show multiple weak peaks that may be related to the thinner thickness of the single layer. Furthermore, it is obvious that the diffraction peaks of the PZT/PZO multilayer thin films are composed of PZT and PZO films, as shown in Figure 2b, which demonstrates the two components of the multilayer films. Moreover, the elongation and widening of the PZT lattice parameter in the multilayer films demonstrate that the PZT film is affected by an in-plane tensile strain from the PZO layer with a larger lattice constant. The cross-sectional images of the PZT and PZO films, and the PZT/PZO multilayer structure are shown in Figure 2c–e. We found that the thickness of the prepared film samples was approximately 500 nm, and the microstructure was dense without pores. The feature of the columnar growth can be seen clearly, and there is a particularly obvious interface between the film and the bottom electrode substrate.

Polarization, as a function of electric field (*P-E*) hysteresis loops of the PZT and PZO films, and the PZT/PZO multilayers tested at 1 kHz, is given in Figure 3. It can be observed that all three thin films exhibit comprehensive performance hysteresis loops. Typical antiferroelectrics behavior can be found in the PZO film, while the PZT and the PZT/PZO multilayers show common FE loops. Nevertheless, the PZT/PZO multilayer films show double *P-E* loops at the weak external electric field. One should also note the higher apportionment of applied *E* on the PZO layer to the lower dielectric constant at the low external electric field. With the increase in the external electric field, the electric field applied on the PZT increases, the PZT layer is obviously polarized, and the *P-E* loop exhibits a normal FE hysteresis loop. For the PZT/PZO multilayer films, the saturation polarization is close to that of PZT, and the residual polarization is much smaller, which enhances the energy-storage properties of the PZT/PZO multilayers with a slender hysteresis loop. Meanwhile, compared with the PZO film, the much higher saturation polarization of the PZT/PZO multilayer films also makes them more attractive.

In order to show the significance of AFE behavior in three different composition films, a change in switching current with an external electric field was also explored (Figure 3d). The four typical switching current peaks of the PZO film correspond to the polarized antiferroelectric to ferroelectric phase, with forward and backward switching fields [22]. Thus, it is noteworthy that a sharp phase transition occurred in the PZO and PZT thin films, yet a diffused transition was also noticed in the PZT/PZO multilayers. The PZT/PZO multilayer thin films have more slanted loops, which may be caused by the contribution of both PZT and PZO to this process [23,24]. Furthermore, the diffused phase transition also brings about a lower phase transition field in the PZT/PZO multilayer films than that found in the PZO thin film; this might be attributable to the dispensation of the applied E between the PZT and PZO layers. It also means that the transformation from sharp phase transition to diffused phase transition can be tuned by the layer structure.

The *E_BDS_* of the three samples was calculated by using the point counting method of Weibull distribution in Figure 4a. The breakdown electric field plays an indispensable part in improving energy-storage density [25]. It should be noted that the electric breakdown strength is 1162, 1373, and 1760 kV/cm, for the PZT, PZO, and PZT/PZO multilayer films, respectively. The enhancement in *E_BDS_* may owe itself to interface resistance and the barrier effect of the interface layer [19]. The interface between the ferroelectric and antiferroelectric layers in the PZT/PZO multilayer structure can effectively prevent the spread of an electric tree, whereas the interlayer charge coupling of the multilayer structure can further increase the *E_BDS_* [16]. It is well understood that the intrinsic effects originating from surface properties, grain boundary, and domain walls have a major influence on the leakage current, thus affecting the dielectric breakdown strength [26,27]. The electric-field-induced current is shown in Figure 4b–d. The time gap was 500 ms at each voltage step in our measurement. The current density of the PZT film is the highest among all the films, and it arrives 7.6 × 10^−4^ A/cm^2^ under the utmost external *E* of 784.5 kV/cm, while the other two samples are relatively low. For the PZO film, current density had an obvious current peak around 300 kV/cm, which nearly correlated to the phase switching field. As the leakage current usually increases monotonically with the voltage, the current peak corresponding to negative resistance can be attributed to the domination of a displacive current, which itself is produced by the phase transition from antiferroelectric to ferroelectric; these findings are consistent with those in previous reports [28,29]. For ferroelectric materials, Krupanidhi et al. pointed out that the ferroelectric polarization current can be ignored in relation to a time gap that is much larger than the domain switching speed [30]. Therefore, the current density of the PZT and PZT/PZO multilayer films was mainly caused by the leakage current. Leakage current behavior is usually analyzed by mechanisms such as the space-charge-limited current (SCLC), Schottky emission (SE), and Poole–Frenkel (P–F) emission models [31,32,33]. Interestingly, the PZT/PZO multilayer thin films show interface-limited SE behaviors in which the log(*I*) is linear with *E*^0.5^ for constant temperature [34]. For the PZT single layer films, the log(*I*) is almost linear with log(*V*) in one or more sections, implying that its leakage currents are dominated by space-charge-limited bulk conduction (SCLC) [34,35,36]. Our study suggests that the layer structure can affect the polarization charge and result in the modulation of the leakage current from SCLC for the PZT single layer films to the SE model for the multilayer thin films.

Given the polarization characteristics and the *E_BDS_*, the energy-storage characteristics of the films were calculated using Formulas (1) and (2), as shown in Figure 5. Obviously, the *W_rec_* of all the samples increases with the increase in external E. The maximum values of 8.12, 15.32, and 21.11 J/cm^3^ were obtained from the PZT, PZO, and PZT/PZO multilayer films, respectively. The *W_rec_* of 21.11 J/cm^3^ for the PZT/PZO multilayers is primarily due to the large electric breakdown field. This also proves that it is a feasible method to increase the *E_BDS_* of the films to delay the saturation polarization by utilizing a multilayer structure to enhance the energy-storage density. The polarization behavior and energy loss will have an extraordinary impact on the energy-storage efficiency of ferroelectrics. Usually, the slender *P-E* loop attains an excellent energy-storage efficiency. In our work, the maximum values are 30.3, 76.0, and 63.3% for the PZT, PZO, and PZT/PZO multilayer films, respectively. It can also be observed that the energy-storage efficiency of the PZT/PZO films is obviously larger than that of the PZT film without sacrificing the polarization. As the *P_m_*–*P_r_* value reduces, the hysteresis loop of the PZT/PZO films becomes slender, and the energy-storage efficiency changes.

The dielectric constant and loss values of the three different samples are strongly dependent on the external *E*, as shown in Figure 6a–c. The PZT thin film shows an obvious ferroelectric butterfly curve, while the PZO thin film shows antiferroelectric double butterfly curves. The PZT/PZO multilayers exhibit dispersed antiferroelectric double butterfly curves in a weak external electric field range. For the PZT/PZO multilayer thin films, the phase transition field is much smaller than that of the single layer PZO film because the major external voltage is applied to the PZO layer as its low dielectric constant in a low external voltage. Figure 6d shows the dielectric features of all the films from low frequency to high frequency at room temperature. The dielectric constant is 950, 288, and 561 (at 1 kHz) for the PZT, PZO, and PZT/PZO multilayer films, respectively. The dielectric constant of the PZT/PZO multilayer films is approximately twice that of PZO, which is due to the existence of PZT. According to the distribution of permittivity, the combination of high and low permittivity materials will make the overall permittivity of the composite multilayer films lower than that of low permittivity materials [37]. However, the experimental results show the opposite situation, which is due to the interfacial polarization between the PZT ferroelectric layer and the PZO antiferroelectric layer [19]. As can be seen from Figure 6a–c, the dielectric loss in all the films is relatively low, less than 0.07, which denotes the high quality of these films.

## 4. Conclusions

In summary, by taking advantage of ferroelectrics with a high *P_s_* and antiferroelectrics with a low *P_r_* and a high *E_BDS_*, a multilayer PZT/PZO thin film was prepared via the sol–gel technique. The voltage strength and, thus, the energy-storage density are raised via the adoption of a multilayer structure, which efficiently hinders the extension of the electric tree. Interestingly, the energy-storage density (*W_rec_*) is 21.11 J/cm^3^ in the PZT/PZO multilayer thin films, which is larger than in the PZT and PZO thin films. The multilayer structure could modulate the current mechanism from space-charge-limited bulk conduction (SCLC) for the PZT and PZO single layer films to Schottky emission (SE) for the PZT/PZO multilayer thin films. All the results demonstrate that the ferroelectric/antiferroelectric multilayer structure films shown in this paper have enormous potential for energy-storage capacitors.

## Figures and Tables

**Figure 1 nanomaterials-11-02141-f001:**
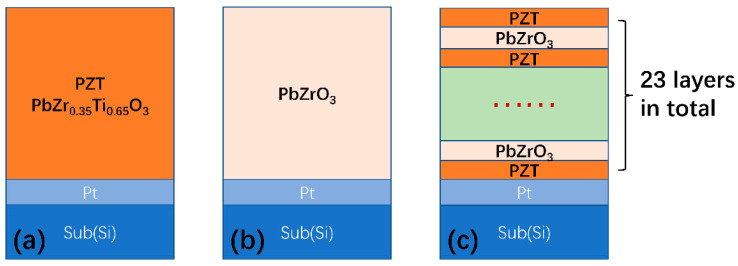
Schematic diagrams of the (**a**) PZT film, the (**b**) PZO film, and the (**c**) PZT/PZO multilayers, respectively.

**Figure 2 nanomaterials-11-02141-f002:**
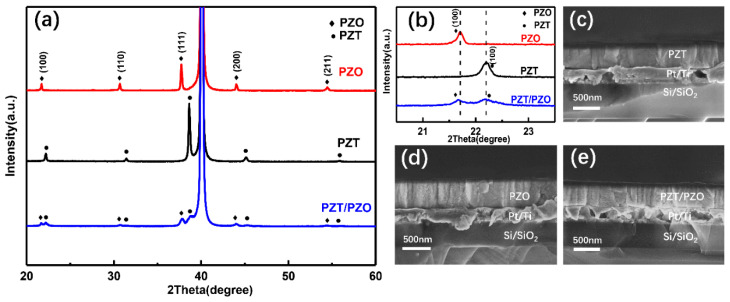
(**a**) XRD patterns of PZT, PZO, and PZT/PZO multilayer films grown on Pt(111)/Ti/SiO2/Si with linear y-scale at room temperature; (**b**) magnified XRD patterns around 2*θ* = 22; cross-section images of (**c**) PZT, (**d**) PZO, and (**e**) PZT/PZO multilayer films.

**Figure 3 nanomaterials-11-02141-f003:**
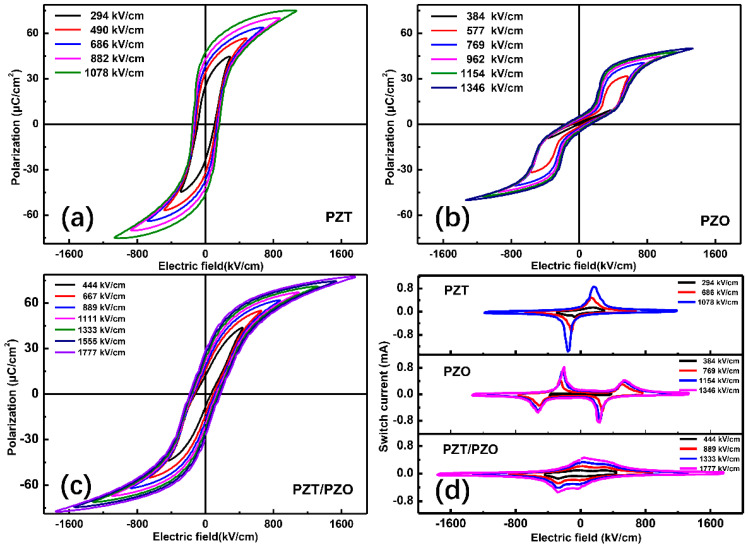
(**a**–**c**) Hysteresis loops of PZT, PZO, and PZT/PZO multilayer films; (**d**) the switch current with the electric fields of PZT, PZO, and PZT/PZO multilayer films at room temperature.

**Figure 4 nanomaterials-11-02141-f004:**
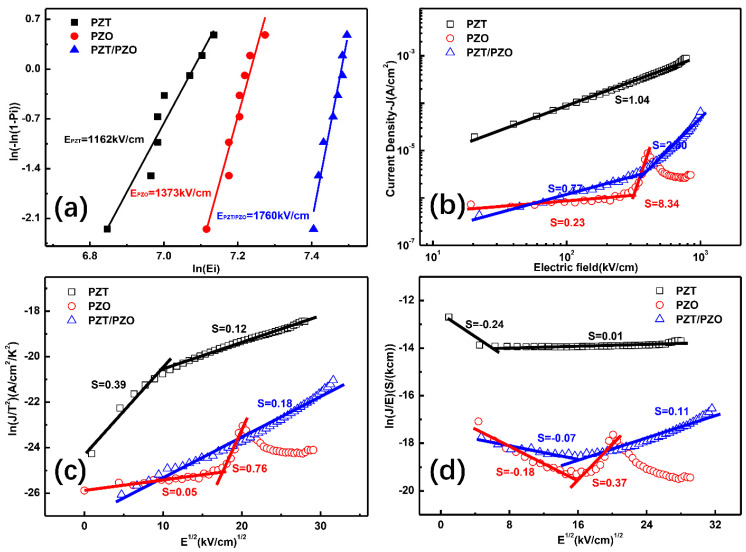
(**a**) Weibull plot of the breakdown electric field *E_BDS_*; (**b**) analysis of current density behavior for PZT, PZO, and PZT/PZO multilayer films; space-charge-limited bulk conduction; (**c**) Schottky emission; (**d**) Poole-Frenkel emission.

**Figure 5 nanomaterials-11-02141-f005:**
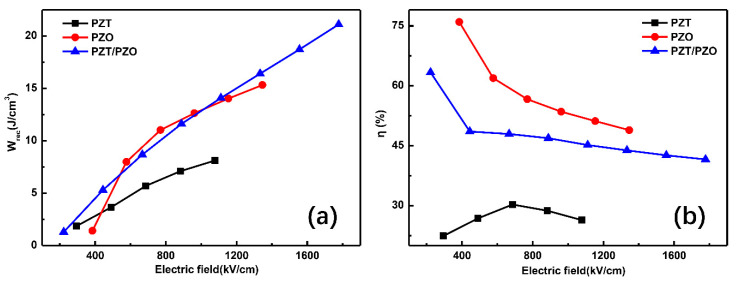
(**a**) The recoverable energy-storage density *W_rec_*; (**b**) energy-storage efficiency η of PZT, PZO, and PZT/PZO multilayer films, as measured at the different external electric fields.

**Figure 6 nanomaterials-11-02141-f006:**
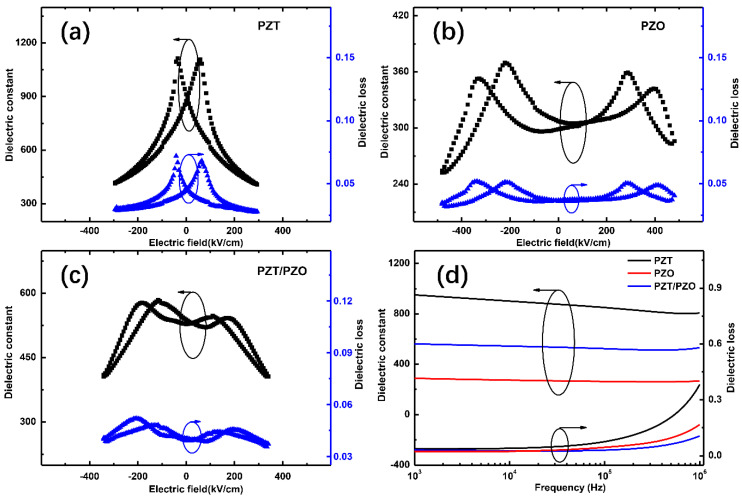
Dielectric properties of (**a**) PZT, (**b**) PZO, and (**c**) PZT/PZO multilayer films as a function of DC electric field at 1 kHz; (**d**) frequency dependence dielectric properties from 1 to 1000 kHz at room temperature.

## Data Availability

The data presented in this study are available on request from the corresponding author.

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
