# Peer review of "Enhancement of Energy-Storage Density in PZT/PZO-Based Multilayer Ferroelectric Thin Films"

_nanomaterials, 2021, doi:10.3390/nano11082141_

Round 1

Reviewer 1 Report

I have carefully read the manuscript entitled "Enhancement Energy-Storage Density in PZT/PZO based multilayer ferroelectric thin films". The authors present structural and ferroelectric characterisation of three films: a PZO film, a PZT film, and a PZT/PZO multilayer. The authors analyse the results in terms on energy-storage parameters. The results are interesting. The manuscript fails in comparing the results with the state of the art. In fact, some important references are missing, which makes the whole work misleading. The manuscript writing is overselling without being that supported by the data. In addition, the "English language" must be revised. I addition, some conclusions are not fully supported by the performed analysis. Overall, I do not recommend to publish the present work. 

Some comments about the text: 

Replace SCLC, SE, EBDS  by its not abbrevited form in the abstract. 
Remove "(Wrec)" in the abstract. 
Remove "new idea" from the abstract. 
L 46: The ")" must not be a subindex
L 50 remove the hyphen
L 56 "delayed polarization" meaning is unclear. Rephrase it. 
L 58 Remove "up"
L 141 Replace improvement by "increase"- 
L186-188. The authors leakage analysis is not detailed. In fact it seems very preliminar according to typical analysis reported in literature to disclose physical origin of leakage current. The conclusions in this regard must be softened. 
L196 Remove "outstanding"
L 215 Remove "discovered to be"
L231 remove "withstand"

Figure 1. replace floors by layers. 
Figure 2a,b. It must be indicated if the y-scale is logarithmic 
Figure 2c,d. Each layer must be clearly identified in the figure.- 
Figure 2. For completeness, it is necessary to show the TEM cross section of PZO film. 
Figure 3a. Include the minor ticks as in the b,c panels and make all the x-scale and y scale equal to allow better comparison. 
Figure 4b. The authors show negative derivative of J versus E for PZO. This is not expected as it would correspond to negative resistance. the authors must clarify this issue. In L177-179 the authors mention that the peak might be a result of the phase induced phase transition. If yes, this means that the current measured in PZO contains a displacive current contribution. In a leakage measurement displacive current contributions can't be present, because the leakage is by definition the current flowing under DC field. 

Without being exhaustive:
In ref 10.1039/C8QI00487K  and 10.1016/j.ceramint.2019.06.266 Wrec=31.96 and 29.7 Jcm-3 are reported, respectively. Both higher than authors submitted results. Therefore the manuscript must be rewritten to eliminate the statements that might be misleading regarding the message that the reader can take about the use of multilayer as a "new" method to obtain high Wrec and the fact that the reported results are the best obtained up to now in multilayered thin films- 

Reference https://doi.org/10.1002/adfm.201807196 is also relevant . 

Similarly, more detailed comparison of the EBDS values with those reported in the literature must be done. 

Author Response

Thanks a lot for your careful reading of the manuscript and providing precious comments and suggestions. According to the kindly suggestion of the reviewer, we list the specific modification, add some explanations and comparing the results with the state of the art.

Reviewer 2 Report

The article investigates PbZr0.35Ti0.65O3 (PZT), PbZrO3 (PZO) films, and PZT/PZO ferroelectric/antiferroelectric mul-tilayer films were deposited on a Pt(111)/Ti/SiO2/Si substrate by sol-gel process technique. It is shown that with the help of the applied technologies the coercive field and recoverable energy storage are increased. Experimental methods are adequate. I recommend this article for publication

Author Response

Thanks a lot for your careful reading of the manuscript and providing precious comments. 

Round 2

Reviewer 1 Report

I have carefully read the revised version of "Enhancement Energy-Storage Density in PZT/PZO based multilayer ferroelectric thin films". The authors have replied my comments. I do not feel that the comparison with the literature is clear enough. However, at this point, reader will have to judge about that. 

I recommend to revise again the overselling statements. Even in the revised sentences the authors still use "great improvement", "improved", "good", etc. This is in my opinion annoying for the reader. 

Author Response

Response Letter(Please see the attachment.)

1.  I do not feel that the comparison with the literature is clear enough. However, at this point, reader will have to judge about that.

Answer: I appreciate the reviewer’s suggestion. We revised and supplemented the unclear part in the original text.

“The above multilayer structures combined with the ferroelectric and dielectric materials have received extensive attention due to the enhancement of electric breakdown strength.”

“The antiferroelectric material is regarded as one of the most promising energy storage materials due to the double hysteresis loops with low remnant polarization.”

“The Wrec is 21.11 J/cm3 at 1760 kV/cm for the PZT/PZO multilayers. Compared with the energy storage density reported in the literature at the same level of operation voltage, such as 14.8 J/cm3 at 1592 kV/cm for PLZT/PZO multilayers and 13 J/cm3 at 2400 kV/cm for PZT/Al2O3/PZT films, our energy storage density is a little higher under similar operation electric field though our maximum energy storage density is not larger than others [7,14,20]. It demonstrates that we also need to enhance the electric breakdown strength in the future.”

2. I recommend to revise again the overselling statements. Even in the revised sentences the authors still use "great improvement", "improved", "good", etc. This is in my opinion annoying for the reader. 

Answer: Thanks a lot for your careful reading of the manuscript and providing precious suggestions. According to your kindly suggestion, we have revised the unreasonable expression in the text.

“The electric breakdown strength of PZT/PZO multilayer structure can be further enhanced to 1760 kV/cm, which is higher than PZT (1162 kV/cm) and PZO (1373 kV/cm) films.”

“Our results demonstrate the multilayer structure is an effective method to enhance the energy storage capacitors.”

“Our results demonstrate that the EBDS value is 1760 kV/cm for PZT/PZO multilayer films, which is much larger than PZT film (EBDS=1162 kV/cm).”

“It can be observed that the energy storage efficiency of PZT/PZO films is obviously larger than that of PZT film without sacrificing the polarization. As the Pm-Pr value is reduced, the hysteresis loop of PZT/PZO films becomes slender, and the energy storage efficiency is changed.”

“In summary, by taking advantage of ferroelectrics with high Ps and antiferroelectrics with low Pr and high EBDS, a multilayer PZT/PZO thin film was prepared via the sol-gel technique.”

“Interestingly, the energy storage density (Wrec) of 21.11 J /cm3 in PZT/PZO multilayer thin films, which is larger than PZT and PZO thin films.”

3.For the editing of English language and style, we have signed the English Language editing service to make sure the high quality of this manuscript (ID: english-33434).